# Test-Retest Reliability and Minimal Detectable Changes for Wearable Sensor-Derived Gait Stability, Symmetry, and Smoothness in Individuals with Severe Traumatic Brain Injury

**DOI:** 10.3390/s25061764

**Published:** 2025-03-12

**Authors:** Fulvio Dal Farra, Stefano Filippo Castiglia, Maria Gabriella Buzzi, Paolo Brasiliano, Sara De Angelis, Gianluca Paolocci, Simona Vasta, Gabriele Marangon, Amaranta Soledad Orejel Bustos, Elena Bergamini, Viviana Betti, Marco Tramontano

**Affiliations:** 1Department Information Engineering, University of Brescia, 25121 Brescia, Italy; fulvio.dalfarra@unibs.it; 2Department of Medico-Surgical Sciences and Biotechnologies, Sapienza University of Rome Polo Pontino, 04100 Latina, Italy; 3IRCCS Fondazione Santa Lucia, via Ardeatina, 354, 00179 Rome, Italy; mg.buzzi@hsantalucia.it (M.G.B.); s.deangelis@hsantalucia.it (S.D.A.); gpaolocci@unime.it (G.P.); a.orejel@hsantalucia.it (A.S.O.B.); viviana.betti@uniroma1.it (V.B.); 4Department of Movement, Human and Health Sciences, University of Rome “Foro Italico”, Piazza Lauro De Bosis 6, 00135 Rome, Italy; paolo.brasiliano@uniroma4.it (P.B.); elena.bergamini@unibg.it (E.B.); 5Department of Biomedical and Dental Sciences and Morphofunctional Imaging, University of Messina, 98125 Messina, Italy; 6Department of Cognitive, Psychological Pedagogical and Cultural Studies (COSPECS), University of Messina, 98122 Messina, Italy; simona.vasta@studenti.unime.it; 7Department of Neuroscience, Imaging and Clinical Sciences, University of Chieti-Pescara, 66100 Chieti, Italy; gabriele.marangon@unich.it; 8Department of Management, Information and Production Engineering, University of Bergamo, Via Marconi 4, 24044 Dalmine, Bergamo, Italy; 9Department of Psychology, Sapienza University of Rome, via dei Marsi 78, 00185 Rome, Italy; 10Department of Biomedical and Neuromotor Sciences, University of Bologna, 40138 Bologna, Italy; marco.tramontano@unibo.it; 11Unit of Occupational Medicine, IRCCS Azienda Ospedaliero-Universitaria di Bologna, 40138 Bologna, Italy

**Keywords:** inertial sensor, minimal detectable change, traumatic brain injury, gait

## Abstract

Severe traumatic brain injury (sTBI) often results in significant impairments in gait stability, symmetry, and smoothness. Inertial measurement units (IMUs) have emerged as powerful tools to quantify these aspects of gait, but their clinometric properties in sTBI populations remain underexplored. This study aimed to assess the test-retest reliability and minimal detectable change (MDC) of three IMU-derived indices—normalized Root Mean Square (nRMS), improved Harmonic Ratio (iHR), and Log Dimensionless Jerk (LDLJ)—during a 10 m walking test for sTBI survivors. Forty-nine participants with sTBI completed the walking test, with IMUs placed on key body segments to capture accelerations and angular velocities. Test-retest analyses revealed moderate to excellent reliability for nRMS and iHR in anteroposterior (ICC: 0.78–0.95 and 0.94, respectively) and craniocaudal directions (ICC: 0.95), with small MDC values, supporting their clinical applicability (MDC: 0.04–0.3). However, iHR in the mediolateral direction exhibited greater variability (ICC: 0.80; MDC: 9.74), highlighting potential sensitivity challenges. LDLJ metrics showed moderate reliability (ICC: 0.57–0.77) and higher MDC values (0.55–0.75), suggesting the need for further validation. These findings underscore the reliability and sensitivity of specific IMU-derived indices in detecting meaningful gait changes in sTBI survivors, paving the way for refined assessments and monitoring the rehabilitation process of sTBI survivors. Future research should explore these indices’ responsiveness to interventions and their correlation with functional outcomes.

## 1. Introduction

Traumatic Brain Injury (TBI) is a leading cause of disability worldwide, with approximately 69 million cases reported annually [1]. The impact of TBI is particularly severe in low- and middle-income countries, where road accidents remain a primary contributor [2]. In high-income countries, falls, sports injuries, and assaults are also significant contributors [3]. Severe TBI (sTBI), defined by a Glasgow Coma Scale (GCS) score of 8 or lower and a coma lasting at least 24 h, often results in long-term cognitive and physical impairments. These effects not only diminish patients’ quality of life but also place a significant socioeconomic burden on healthcare systems [4].

Rehabilitation of sTBI patients requires a multidisciplinary approach to enhance recovery and functional independence [5]. Key strategies include physiotherapy for mobility and balance, occupational therapy for daily activities, and cognitive rehabilitation therapy [6]. Emerging interventions, such as virtual reality and robotic-assisted therapies, are gaining attention for their potential benefits [7,8]. However, due to the heterogeneity of TBI presentations, personalized treatment programs remain essential, emphasizing the need for accurate assessment tools [7,8,9]. Postural and gait stability and motor impairments are critical challenges in sTBI recovery [10,11,12,13,14,15], underscoring the importance of precise assessments to identify limitations and monitor balance and gait rehabilitation effectiveness [9].

Wearable inertial measurement units (IMUs) have enhanced gait assessment by providing objective, portable, and accessible alternatives to traditional methods [16,17]. Depending on the number of sensors and their placement setup, IMUs, which embed accelerometers, gyroscopes, and magnetometers, are used in gait analysis to retrieve several kinematic and spatio-temporal gait parameters. Moreover, by directly analyzing the acquired signals during gait, IMUs offer insights into stability, symmetry, and movement smoothness. Clinically relevant metrics, such as normalized Root Mean Square (nRMS) for stability, improved Harmonic Ratio (iHR) for symmetry, and Log Dimensionless Jerk (LDLJ) for smoothness, have demonstrated clinical relevance in sTBI gait assessment [9,10,11,18]. Their integration with machine learning models further enhances diagnostic precision [19,20,21].

Despite the increasing adoption of IMU-derived gait metrics, their reliability and Minimal Detectable Change (MDC) require further validation. MDC, which represents the smallest measurable change beyond random variation, is crucial for result interpretation and optimizing rehabilitation strategies [22,23]. While nRMS, iHR, and LDLJ have been studied in neurological populations [24,25,26], research on their test-retest reliability and MDC in sTBI remains scarce [27]. Addressing this gap is essential for advancing evidence-based clinical decision-making, improving sensitivity in tracking rehabilitation progress, and standardizing wearable sensor-based gait assessments [28]. Furthermore, when calculating IMU-derived indices for linear walking at steady state, a time series of at least 20 strides is typically required [26,29,30]. However, this requirement implies a large acquisition space, which is often unavailable in clinical settings, restricting the application of these metrics outside of laboratory contexts.

We hypothesized that IMU-derived gait indexes would result in reliable and sensitive change scores even for shorter gait bouts, such as during a 10 m walking test (10MWT), with acceptable intra-class correlation coefficients (ICC) and MDC values. This would enhance the nuance of disposable gait assessment tools for monitoring the rehabilitation process of sTBI survivors. Therefore, this study aims to calculate the test-retest reliability and the MDC of the nRMS, iHR, and LDLJ parameters, measured through IMU sensors, in survivors from sTBI.

## 2. Materials and Methods

This cross-sectional study was carried out at Santa Lucia Hospital (Institute for Research and Healthcare) from January 2020 to December 2023 and was approved by the Local Independent Ethics Committee with protocol number CE/PROG.803.

### 2.1. Inclusion Criteria

All participants were included in the study after providing their informed consent and thorough evaluation of the inclusion and exclusion criteria. Inclusion criteria were as follows: age between 15 and 65 years; GCS score ≤ 8 (used to objectively describe the severity of impaired consciousness at the time of injury) [31]; Level of Cognitive Functioning (LCF) ≥ 7 [32]; ability to understand verbal commands; and ability to walk without any device or need for continuous physical assistance (Functional Ambulation Classification > 3). The exclusion criteria were cognitive deficits affecting the capacity of a person to understand the task instructions (Mini-Mental State Examination score > 24) [33], severe unilateral spatial neglect, severe aphasia, and the presence of other neurological conditions, orthopedic comorbidities, or cardiac comorbidities. All procedures comply with the ethical standards of the relevant national and institutional guidelines on human experimentation and with the World Medical Association Declaration of Helsinki. The reporting of this study is based on the “Strengthening the Reporting of Observational Studies in Epidemiology” (STROBE) [34].

### 2.2. Data Collection

To ensure high-quality data collection, raters underwent specialized training in administering clinical outcome measures and performing kinematic assessments. Inertial sensors were securely attached to specific body segments using Velcro straps to reduce oscillations and minimize motion artifacts. Participants were asked to walk at a comfortable pace along a 10 m straight path with 2 m auxiliary sections at each end, for a total walking distance of 14 m [35]. To investigate natural and spontaneous locomotion, participants were provided with general, qualitative instructions, allowing them to freely choose their gait speed without any external sensory input. The hallway had no visible pavement connections or border lines, and indirect lighting was equally distributed along its length. There were no adverse events recorded during the procedures. During assessments, two physiotherapists remained close to the participants to prevent falls and ensure proper execution of the tests [18]. This procedure was repeated twice within 10 min using an identical experimental setup, serving as a test-retest paradigm. Specifically, the first and second acquisitions were considered for the analysis [9,18] (Figure 1).

Instrumental evaluations were carried out by two physiotherapists with specialized expertise in gait analysis using IMUs. During the trials, participants were equipped with five synchronized IMUs (128 Hz, Opal, APDM, Portland, OR, USA), capturing three-dimensional linear accelerations and angular velocities with ranges of ±16 g and ± 2000°/s, respectively. The IMUs were positioned as follows: on the occipital bone near the lambdoid suture (head, H), at the sternum’s center (S), at the L4/L5 vertebral level above the pelvis (P), and on each leg just above the lateral malleoli for step and stride segmentation [36] (Figure 1). Before each acquisition session, a calibration procedure was implemented to ensure consistent sensor alignment between test sessions using the “Motion Studio” software (Version 1.0 Copyright 2017, APDM, Inc., Portland, OR, USA). In the static phase at the beginning of each trial (from 3 to 5 s), the gyroscope static bias was removed. Then, to guarantee identical starting conditions for all IMUs located on the upper body, a reference system aligned with the gravity vector was established for each IMU using acceleration data. The rotational matrix between each IMU and this reference system was calculated and applied to the dynamic phase, aligning accelerometer and gyroscope data to approximate the anatomical axes: anteroposterior (AP), mediolateral (ML), and craniocaudal (CC), with the latter referring to the longitudinal axis. Gravity was then removed from the CC axis of accelerometer data [37]. To filter the data, a second-order Butterworth low-pass filter was applied with cutoff frequencies of 10 Hz for accelerometers and 6 Hz for gyroscopes. Data processing was performed using MATLAB^®^ (MATLAB R2022b, The MathWorks Inc., Natick, MA, USA) to extract acceleration-derived metrics, including nRMS, iHR, and LDLJ. These metrics are commonly referenced in the neurorehabilitation literature for assessing stability, symmetry, and gait smoothness, though their reliability remains under investigation [38].

### 2.3. Sensor-Derived Indexes

nRMS was calculated using the acceleration signals at the pelvis, trunk, and head levels. The RMS values for each stride for the AP, ML, and CC components were first calculated. Then, the AP and ML components were normalized by dividing them by the CC component to account for inter-individual differences in overall acceleration magnitudes caused by variations in walking speed. Higher nRMS values indicated greater acceleration and reduced stability [39,40]. The formula is as follows:(1)$$nRMS=\frac{\sqrt{\frac{\Sigma_{i}^{N}=1^{x^{2}}}{N}}}{\sqrt{\frac{\Sigma_{i}^{N}=1^{y^{2}}}{N}}}$$
where *x* is the AP or ML components, *y* is the CC component of the acceleration signal, and *N* is the number of samples of each stride.

The improved Harmonic Ratio, iHR, was calculated for each acceleration signal direction acquired at the pelvis level. After subtracting the mean value from each acceleration component, the iHR was calculated for each stride as the ratio of the power amplitudes of the k intrinsic harmonics to the total power amplitude of the signal, considering the first 20 harmonics. iHR values range from 0% (complete asymmetry) to 100% (perfect symmetry of the acceleration signals). iHR was calculated as follows [41]:(2)$$iHRj=\frac{\Sigma_{i=1}^{K}P_{I}^{i}}{\Sigma_{i=1}^{K}\left(P_{I}^{i}+P_{E}^{i}\right)}\times 100$$

The Log Dimensionless Jerk, LDLJ, was calculated at the pelvis level using linear acceleration and angular velocity data for each spatial direction, producing two indices: LDLJ(a) for acceleration and LDLJ(w) for angular velocity. Lower LDLJ values indicate smoother movement [42]. LDLJ was calculated as follows:(3)$${LDLJ}_{s\;}=-\ln\left(\frac{{\left(t_{2}-t_{1}\right)}^{3}}{max\left(\|s(t){\|}_{2}^{2}\right)}\;\int_{t_{1}}^{t_{2}}\|\frac{d}{dt}\;s(t){∥}_{2}^{2}dt\right)$$
where *s*(*t*) represents the linear accelerations or angular velocity data and *t_1_* and *t_2_* are the starting and ending points, respectively, of the gait cycle.

### 2.4. Clinical Assessment

For each patient, a clinical assessment was performed using the Dynamic Gait Index (DGI) and the Berg Balance Scale (BBS) [43,44,45]. DGI was used to assess a participant’s ability to modify gait in response to changing task demands. It consists of items rated from 0 to 3 (0 = severely impaired; 3 = physiological performance), yielding a maximum score of 24 points. A score lower than 19 points has been associated with impairment of gait and fall risk [43,44]. The BBS was used to determine a patient’s ability (or inability) to safely balance during a series of predetermined tasks. It is a 14-item list with each item consisting of a five-point ordinal scale ranging from 0 to 4, with 0 indicating the lowest level of function and 4 indicating the highest level [45].

### 2.5. Statistical Analysis

Table 1 reports the sample’s demographics and clinical characteristics through descriptive statistics such as mean, standard deviations (SD), range, and percentages. For each acceleration index, the mean and SD were calculated, as well as the difference between test and retest (see Table 2). After checking for the normality of the distributions through the Shapiro—Wilk test, paired samples *t*-test, or Wilcoxon test, Cohen’s effect size (d) and 95% confidence interval were calculated to assess the significance and magnitude of the differences between the gait trials; “d” values around 0.2 suggest a small effect, 0.5 a medium effect, and 0.8 or higher a large effect [46]. To assess the test-retest reliability, the intra-class correlation coefficient (ICC 3;1) was calculated using a two-way mixed-effects model for absolute agreement within its 95% confidence interval (CI). This choice is justified by the fact that the same raters evaluated all participants, and we aimed to assess the consistency of their ratings rather than generalizing the results to other potential raters [47,48]. ICC values were interpreted as follows: less than 0.5 as poor reliability; 0.5–0.75 as moderate reliability; 0.75–0.9 as good reliability; and greater than 0.90 as excellent reliability [47]. Statistical analyses were carried out at a 95% significance level. To calculate the standard error of measurement (SEM), required to calculate the Minimal Detectable Change (MDC), the following formula was used [48]:(4)$$SEM\;=\;SD\;\sqrt{1-ICC}$$

Finally, the MDC was calculated using the following formula:(5)$$MDC=SEM\times 1.96\times \sqrt{2}$$

To assess the impact of age and gender on test-retest reliability, a subgroup analysis was performed using a Mann—Whitney test. No missing data across the variables and observations were found.

## 3. Results

Data from forty-nine participants with sTBI (17 females, aged 36.7 ± 13.2 years, BMI 23.6 kg/m^2^ ± 3.8) were analyzed (see Table 1). The majority of the selected patients sustained sTBIs as a result of traffic accidents, while one patient suffered an sTBI due to a fall from a horse and another due to a fall from a building. The only variable which was not normally distributed was the “months from trauma”.

### 3.1. Test-Retest Reliability

No significant differences were found between test-retest assessments (Table 2).

Test-retest analyses revealed moderate to excellent reliability for most of the analyzed gait indices. The intraclass correlation coefficient (ICC) values for nRMS ranged from 0.78 (nRMS-AP at the head) to 0.96 (nRMS-ML at the pelvis and trunk), indicating good or excellent reliability for the majority of measures. iHR also demonstrated good to excellent reliability, with ICC values ranging from 0.80 (iHR-ML) to 0.95 (iHR-AP and iHR-CC). For LDLJ, reliability showed greater variability, with ICC values ranging from 0.61 (LDLJa-CC) to 0.77 (LDLJw-ML). Descriptive statistics for the acceleration-derived metrics and test-retest differences are reported in Table 2. The distributions of the test-retest differences are illustrated in Figure 2, whereas the ICCs for the nRMS, iHR, and LDLJ parameters are reported in Table 3. No significant differences were found between younger (<40 years) and older (>40 years) participants, nor between male and female subgroups (*p* > 0.05).

### 3.2. Standard Error of Measurement (SEM) and Minimal Detectable Change (MDC)

Regarding nRMS, the lowest MDC was observed for the pelvic and trunk placings, ranging from 0.04 to 0.09, whereas the largest was observed when nRMS was calculated at the head level in ML direction (0.30). For iHR, the lowest MDC was observed for the CC component (2.51), whereas the highest was found for the ML component (9.74). For LDLJ, the lowest MDC was observed for LDLJa-AP (0.55), whereas the highest was reported for LDLJa-CC (0.75). SEM and MDC values for each index are reported in Table 3.

## 4. Discussion

This study aimed to assess the test-retest reliability and the minimal detectable change scores of a set of IMU-derived indices quantifying gait stability, symmetry, and smoothness, in a population of sTBI survivors as assessed during a 10 m walking test. The results showed moderate to excellent test-retest reliability for all investigated parameters, with nRMS and iHR showing particularly high reliability in specific directions. Notably, nRMS calculated from pelvic and trunk AP and ML accelerations demonstrated excellent reliability and low MDC values, indicating its suitability for tracking gait stability changes in this population. The iHR also demonstrated high reliability in the AP and CC directions, but its ML component showed greater variability. In contrast, LDLJa and LDLJw had higher test-retest differences and larger MDC values, indicating greater measurement variability. These findings indicate that, while some IMU-based gait parameters offer solid clinical applicability, others require further refinement to improve reliability in short walking bouts.

Overall, no significant differences were found between test and retest assessments for any parameters investigated (Table 2), demonstrating moderate to excellent reliability (Table 3). Specifically, the nRMS showed excellent test-retest reliability when calculated using both AP and ML pelvic and trunk accelerations. This is consistent with previous research on the reliability of acceleration-derived RMS in healthy adults performing a 10 m walking test [27,49], where ICC values of about 0.80 for RMS were reported [49]. These findings suggest that nRMS can be considered reliable in TBI survivors, even for short walking bouts. Furthermore, both pelvic and trunk nRMS of AP and ML accelerations exhibited small MDCs (Table 2), indicating that minimal changes in nRMS are required to reflect true modifications. As nRMS is widely described as a measure of gait stability [50], these findings contribute to a better understanding of intervention effects on gait stability in sTBI survivors by providing reliable scores that account for true changes beyond measurement error. Although nRMS at the head level in the AP direction showed good ICC values, it demonstrated lower reliability and higher MDC values than measurements from other body levels. This can be attributed to the variability of head motion behavior in the sagittal and transverse plane during gait, which is further altered in sTBI survivors [28,51,52]. Moreover, when using parameters derived from head motion, it is important to consider the patient’s behavior during the test. Factors such as unintended head rotations or flexions may introduce variability, potentially impacting data reliability. For example, nRMS in the craniocaudal direction showed greater variability. This may be affected by unintentional factors, such as voluntary eye movements or compensatory trunk adjustments, potentially reducing measurement reliability. Considering the natural tendency of individuals to make subtle postural adaptations during assessment, these secondary movements may introduce variability that is difficult to control. This highlights the need for strategies to minimize their impact in future studies [50]. Providing patients with clear instructions (e.g., to keep their gaze forward and avoid head movements) could minimize such variability. However, these directives might alter the spontaneous locomotor pattern, potentially reducing the ecological validity of the assessment. Consequently, nRMS should be used cautiously when assessing the effects of rehabilitation on head stability during gait in this population. Additionally, no significant differences were found between younger and older participants, nor between male and female groups. These findings suggest that the reliability of the gait indices is not strongly influenced by demographic factors, reinforcing their robustness for clinical assessments. The iHR, which reflects trunk symmetry [53] during gait, demonstrated excellent reliability in the AP and CC directions, with small MDC values (Table 3). Although achieving acceptable reliability typically requires longer walking bouts (e.g., at least 20 consecutive strides) [29,30], the findings suggest that iHR remains reliable even for shorter gait assessments. Previous studies have shown that iHR is able to identify gait impairments during a 10 m walking test in TBI survivors compared to healthy adults [38]. This study extends its applicability by confirming that iHR in the AP and CC directions is reliable for shorter walking bouts, making it usable for clinical settings. On the other hand, the iHR ML component exhibited larger MDC values, suggesting a reduced sensitivity to slight changes. This is in line with previous research on healthy subjects, where the iHR ML showed substantial between-subject variability [54,55,56]. As shown in Table 1, iHR in the ML direction exhibited the largest confidence intervals, likely due to variability in the clinical conditions of sTBI survivors [52]. Additionally, the high variability in the frequency content of ML trunk motion during gait, exacerbated by the short walking bout, may have contributed to the larger MDC values [55,56]. These findings suggest that while iHR in the ML direction can differentiate sTBI survivors, interventions must produce substantial changes for these differences to be considered significant. As a result, subtle but meaningful changes may go undetected. Further research should explore the ability of iHR to differentiate clinical subgroups of sTBI survivors and assess its responsiveness to rehabilitation interventions.

LDLJa and LDLJw exhibited greater variability in test-retest differences (as reported in Figure 2) and ICC values across the acceleration and angular velocity directions, respectively. This resulted in moderate reliability values and large MDC values relative to LDLJ baseline values (Table 3). Although these indices have been shown to differentiate sTBI survivors from healthy adults [38] and improve after rehabilitation [18], the findings of this study emphasize the need for further investigation to optimize their use and improve the interpretation of changes in gait smoothness during short walking bouts. To the best of our knowledge, this is the first study to quantify the MDC of IMU-derived indices in sTBI survivors.

These results highlight the potential of IMU-based gait assessment in clinical rehabilitation, enabling therapists to monitor recovery progress and tailor interventions based on quantitative gait metrics. Prioritizing stable parameters such as nRMS and iHR may enhance clinical decision-making for sTBI patients by focusing gait rehabilitation programs on gait stability and trunk symmetry

Despite promising results, the following limitations should be acknowledged. While the sample size was sufficient for preliminary reliability estimates, it may not fully capture the clinical heterogeneity of the sTBI population, limiting subgroup analyses. Additionally, the use of a specific evaluation protocol and a single type of sensor may constrain the generalizability of the findings. For example, using only self-selected speed may limit the study’s results’ applicability to walks of varying speeds. The absence of longitudinal data is another limitation; while MDC provides insight into intra-subject variability, future studies should explore its relationship with clinical changes over time. Moreover, other properties, such as responsiveness to interventions and correlations with clinical and instrumental gait metrics, need to be evaluated to validate the practical application of these indices. The findings are based on a controlled clinical environment, limiting generalizability to real-world settings. Factors such as fatigue and cognitive function could also influence gait stability and should be investigated in future studies.

## 5. Conclusions

This study provides useful insights into the reliability and sensitivity to change of IMU-derived indices for monitoring gait stability, symmetry, and smoothness in sTBI survivors. When calculated during a 10 m walking test, the nRMS at the pelvis and upper trunk levels, along with the iHR derived from AP and CC acceleration directions, demonstrated good to excellent test-retest reliability and small MDC values. However, caution is advised when interpreting the nRMS at the head level, the iHR in the ML direction, and the LDLJ metrics, particularly for short walking bouts. These results highlight the potential of IMU-based gait assessment in clinical rehabilitation, enabling therapists to monitor recovery progress and tailor interventions based on quantitative gait metrics. Clinicians can plan gait rehabilitation interventions for TBI survivors that focus on trunk stability and symmetry by assessing RMS and iHR from IMUs. Further research is required to refine these indices and explore their reliability across different populations with gait impairments and under varying acquisition protocols, as well as their responsiveness to rehabilitation.

## Figures and Tables

**Figure 1 sensors-25-01764-f001:**
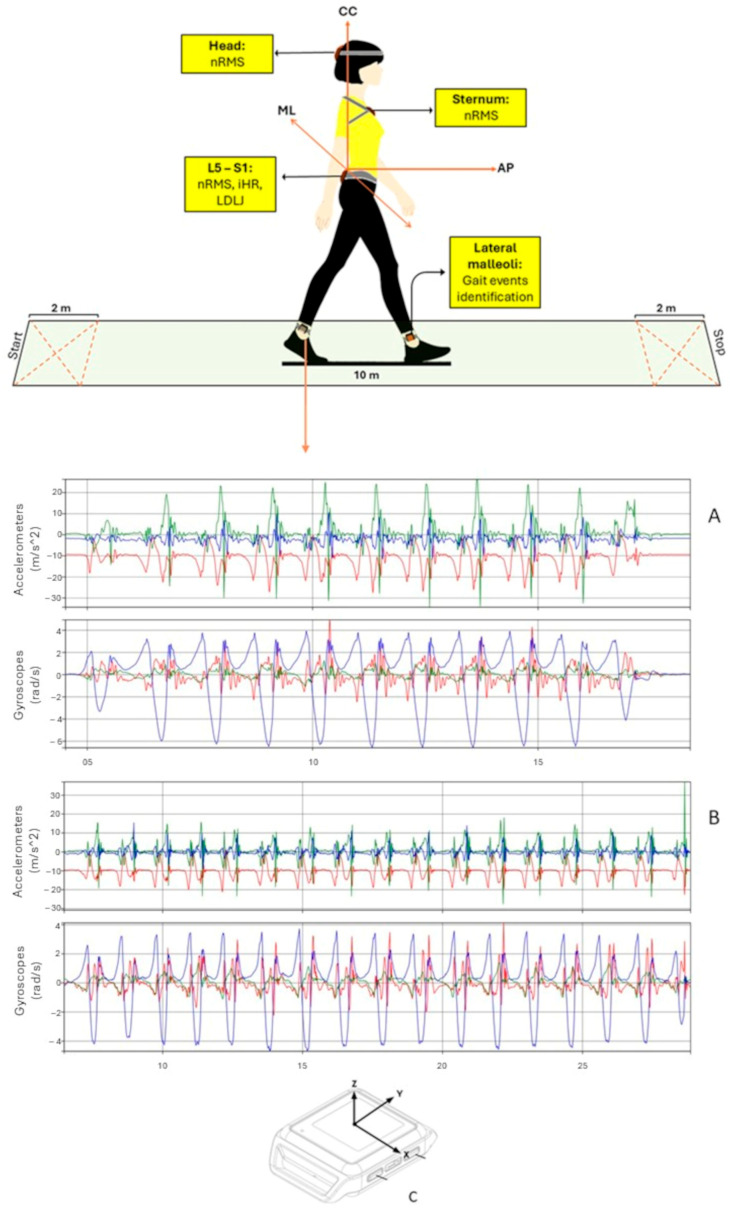
Acquisition protocol and IMUs placement. nRMS, normalized root mean square of the acceleration signals; iHR, improved harmonic ratio; LDLJ, log dimensionless jerk score. CC, ML, AP, craniocaudal, mediolateral, and anteroposterior components of the signals, respectively; (**A**), example of accelerations and angular velocity signals derived from the left leg of a healthy subject; (**B**), example of accelerations and angular velocity signals derived from the left leg of a TBI survivor; (**C**), axes orientation.

**Figure 2 sensors-25-01764-f002:**
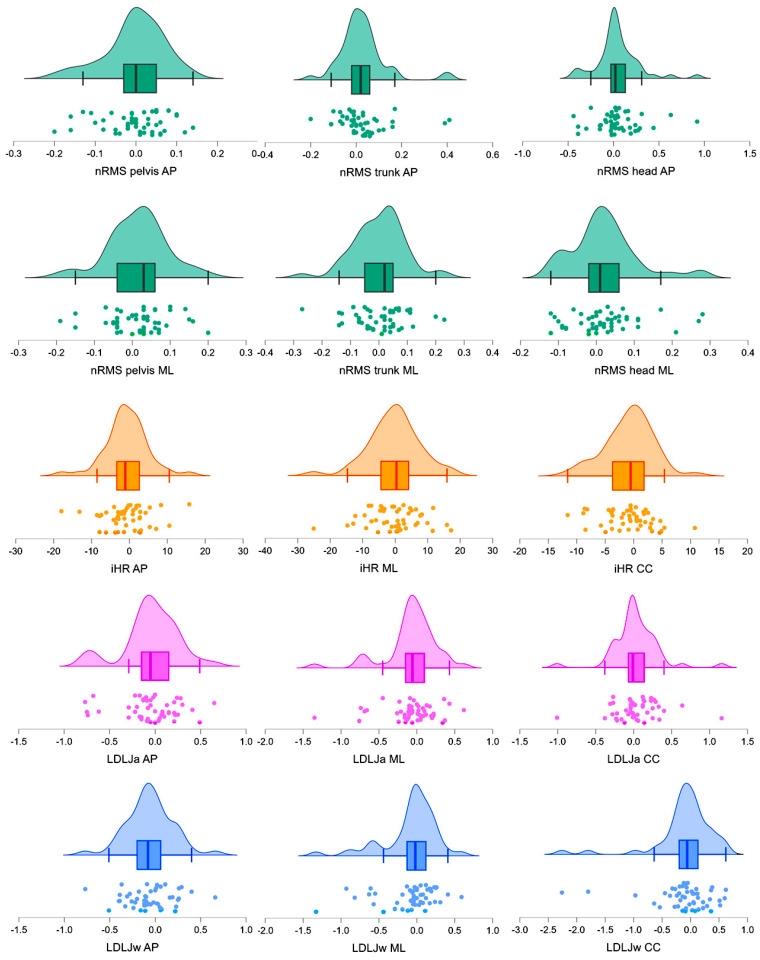
Distributions of the test-retest differences within the sample. Combined density, boxplot, and scatterplot representations of normalized root mean square (nRMS) values, improved harmonic ratios (iHR), and log dimensionless jerk score of accelerations (LDLJa) and angular velocities (LDLJw) across anteroposterior (AP), mediolateral (ML), and craniocaudal (CC) directions. Each plot consists of a density plot, indicating the distribution of values; a boxplot, summarizing the median and interquartile range; and a scatterplot, displaying individual test-retest differences to reflect underlying variability. The x-axes represent the test-retest differences for each respective metric, while the y-axes show the density estimates for the distribution of differences.

**Table 1 sensors-25-01764-t001:** Demographic and clinical characteristics.

	Mean (SD)	Min–Max
Age (years)	36.7 (13.2)	17–67
Sample (F/M)	49 (17/32)	NA
Time since trauma (months)	9 (6.75) *	3–26
Leg length (cm)	76.6 (5.08)	66–85
BMI (kg/m^2^)	23.6 (3.8)	20.7–30.4
Walking speed	0.79 (0.16)	0.5–1.1
Berg Balance Scale (score)	48.4 (7.78)	40–56
Dynamic gait Index (score)	18.96 (4.76)	13–24

* Data expressed in median (IQR) NA = not applicable.

**Table 2 sensors-25-01764-t002:** Descriptive statistics and test-retest difference for each of the assessed index.

Index		Test (95% CI)	Re-Test (95% CI)	Test-Retest Difference (95% CI)	p	d
nRMS	AP (pelvis)	0.82 (0.76, 0.89)	0.83 (0.76, 0.90)	0.05 (0.04, 0.07)	0.85	0.03
	ML (pelvis)	0.85 (0.76, 0.93)	0.83 (0.75, 0.91)	0.06 (0.05, 0.07)	0.17	0.20
	AP (trunk)	0.61 (0.53, 0.69)	0.58 (0.51, 0.65)	0.07 (0.05, 0.09)	0.10	0.27
	ML (trunk)	0.71 (0.62, 0.80)	0.71 (0.62, 0.80)	0.07 (0.05, 0.09)	0.81	0.02
	AP (head)	0.68 (0.57, 0.79)	0.64 (0.55, 0.73)	0.15 (0.10, 0.21)	0.18	0.19
	ML (head)	0.68 (0.59, 0.78)	0.66 (0.57, 0.75)	0.06 (0.05, 0.08)	0.13	0.25
iHR	AP	75.00 (70.07, 79.93)	75.74 (70.77, 80.72)	4.18 (3.07, 5.29)	0.37	0.13
	ML	71.26 (67.75, 74.78)	71.60 (67.96, 75.23)	5.83 (4.34, 7.33)	0.76	0.04
	CC	78.20 (74.07, 82.32)	78.99 (74.96. 83.01)	3.18 (2.37, 4.00)	0.20	0.19
LDLJa	AP	−5.21 (−5.31, −5.11)	−5.17 (−5.26, −5.08)	0.22 (0.16, 0.28)	0.54	0.14
	ML	−5.43 (−5.53, −5.33)	−5.37 (−5.49, −5.24)	0.22 (0.15, 0.30)	0.30	0.20
	CC	−5.06 (−5.15, −4.97)	−5.09 (−5.17, −5.00)	0.19 (0.13, 0.26)	0.50	0.10
LDLJw	AP	−4.57 (−4.72, −4.43)	−4.52 (−4.66, −4.37)	0.20 (0.15, 0.24)	0.11	0.24
	ML	−4.73 (−4.88, −4.58)	−4.66 (−4.81, −4.50)	0.23 (0.16, 0.31)	0.52	0.22
	CC	−4.39 (−4.59, −4.18)	−4.29 (−4.48, −4.10)	0.31 (0.19, 0.43)	0.38	0.19

Abbreviations. nRMS: normalized root mean square; iHR: improved harmonic ratio; LDLJ: log dimensionless jerk; AP: anteroposterior; ML: mediolateral; CC: craniocaudal; a: acceleration; w: angular velocity; p: 95% significance of the test-retest differences; d: Cohen’s effect size.

**Table 3 sensors-25-01764-t003:** Intraclass correlation coefficient, standard error of measurement, and minimal detectable change for each acceleration index.

Index		ICC (95% CI)	SEM (95% CI)	MDC (95% CI)
nRMS	AP (pelvis)	0.95 (0.91–0.97)	0.01 (−0.01, 0.03)	0.04 (−0.04, 0.13)
	ML (pelvis)	0.96 (0.92–0.97)	0.01 (−0.01, 0.03)	0.05 (−0.05, 0.14)
	AP (trunk)	0.92 (0.86–0.95)	0.02 (−0.02, 0.06)	0.09 (−0.09, 0.27)
	ML (trunk)	0.96 (0.93, 0.98)	0.01 (−0.01, 0.03)	0.05 (−0.05, 0.14)
	AP (head)	0.78 (0.65, 0.86)	0.08 (−0.08, 0.24)	0.30 (−0.29, 0.88)
	ML (head)	0.96 (0.93, 0.98)	0.01 (−0.01, 0.03)	0.05 (−0.05, 0.14)
iHR	AP	0.94 (0.90, 0.97)	0.95 (−0.91, 2.81)	3.69 (−3.54, 10.92)
	ML	0.80 (0.68, 0.88)	3.51 (−3.37, 10.39)	9.74 (−9.36, 28.85)
	CC	0.95 (0.92, 0.97)	0.64 (−0.61, 1.89)	2.51 (−2.41, 7.43)
LDLJa	AP	0.57 (0.35, 0.74)	0.19 (−0.18, 0.56)	0.55 (−0.53, 1.62)
	ML	0.63 (0.43, 0.78)	0.20 (−0.19, 0.59)	0.57 (−0.55, 1.7)
	CC	0.61 (0.40, 0.76)	0.27 (−0.26, 0.8)	0.75 (−0.72, 2.23)
LDLJw	AP	0.69 (0.52, 0.81)	0.31 (−0.3, 0.92)	0.68 (−0.65, 2.01)
	ML	0.77 (0.62, 0.86)	0.30 (−0.29, 0.89)	0.57 (−0.55, 1.7)
	CC	0.72 (0.56, 0.83)	0.32 (−0.31, 0.95)	0.65 (−0.62, 1.92)

ICC: intraclass correlation coefficient, SEM: standard error of measurement; MDC: minimal detectable change; 95% CI: 95% confidence interval; nRMS: normalized root mean square; iHR: improved harmonic ratio; LDLJ: log dimensionless jerk; AP: anteroposterior; ML: mediolateral; CC: craniocaudal a: acceleration w: angular velocity.

## Data Availability

The data associated with this paper are not publicly available but are available from the corresponding author on reasonable request.

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
