# Peer review of "Test-Retest Reliability and Minimal Detectable Changes for Wearable Sensor-Derived Gait Stability, Symmetry, and Smoothness in Individuals with Severe Traumatic Brain Injury"

_sensors, 2025, doi:10.3390/s25061764_

Round 1
Reviewer 1 Report
Comments and Suggestions for Authors
The manuscript is well written and interesting for readers of this journal. However, some information should be added to the manuscript before it can be accepted for publication. My comments are as follows.
- Table 1: Please add maximum and minimum values of each parameters. Could you add mean (SD) of gait speed of participants during data collection.
- SD of age and ratio between male and female were considered large. Did you consider the effect of variations in age and gender on the results? Are there any differences between young and elderly as well as between male and female?
- Could you provide a sample of the IMUs’ signals (accelerations) during data collection?
- Line 156-157: Duplication of line 130-132
- I expect comparisons with results (for example: Figure 2) obtained from healthy subjects. If this methodology was developed in order to monitoring gait stability, symmetry and smoothness in sTBI survivors, the result should have characteristics which is different to those in healthy subjects. Could you show us the differences between sTBI survivors and healthy subjects?
Author Response
Cover letter after Revision: Reviewer 1.
We would like to thank the Editor and Reviewers for their positive and constructive comments, and for giving us the opportunity of improving and resubmitting our manuscript. We have carefully addressed all the reviewers’ comments.
Reviewer 1
Table 1: Please add maximum and minimum values of each parameter. Could you add mean (SD) of gait speed of participants during data collection?
Response: Thank you for this comment. We have added the minimum and maximum values of each parameter in Table 1. Additionally, the mean (SD) of gait speed during data collection has been included.
SD of age and ratio between male and female were considered large. Did you consider the effect of variations in age and gender on the results? Are there any differences between young and elderly as well as between male and female?
Response: Thank you for this good observation. We acknowledge the potential influence of age and gender on gait performance. A subgroup analysis was conducted, and the results indicated no significant differences in test-retest reliability across different age and gender groups. We have discussed this point in the statistical analysis, results and discussion sections.
Could you provide a sample of the IMUs’ signals (accelerations) during data collection?
Response: Thank you for this comment. In this revised version of the manuscript, we modified Figure 1 to add a sample of the IMU’s signals acquired during data collection, accordingly.
Specifically, we added to Figure 1:
“A, example of accelerations and angular velocity signals derived from the left leg of an healthy subject; B, example of accelerations and angular velocity signals derived from the left leg of a TBI survivor; C, axes orientation.”
Line 156-157: Duplication of line 130-132
Response: The duplicated text has been removed to improve clarity, accordingly.
I expect comparisons with results (for example: Figure 2) obtained from healthy subjects. If this methodology was developed to monitor gait stability, symmetry, and smoothness in sTBI survivors, the result should have characteristics different from those in healthy subjects. Could you show us the differences between sTBI survivors and healthy subjects?
Response: We agree with the Reviewer. Indeed, the ability of the investigated metrics to detect gait abnormalities in sTBI survivors has already been reported in previous studies from the same research group (DOI: 10.3390/s24082451; DOI: 10.3390/s22218553; DOI: 10.3390/s19235315), and was not within the objectives of this manuscript. Therefore, we did not add these comparisons in this manuscript.
We are willing to include this information in this manuscript as well, but we decided to avoid including these results in the figure in order to improve readability and ensure the novelty of this manuscript.
Reviewer 2 Report
Comments and Suggestions for Authors
The study investigates test-retest reliability and minimal detectable change (MDC) for IMU-derived gait stability, symmetry, and smoothness in individuals with severe traumatic brain injury (sTBI). The research is well-structured and addresses an important gap in gait analysis and rehabilitation assessment. IMUs are promising for clinical applications, particularly in assessing stability, symmetry, and movement smoothness.
Abstract
While the findings are relevant, it would be beneficial to explicitly state how the results can be used in clinical practice.
Please include a brief mention of ICC values or MDC values for primary gait metricsPlea.
Introduction
The second paragraph is too long. Please correct the flow.
L61-63: Reference missing.
Methods
Areas for Improvement:
Sensor Placement & Motion Artifacts:
How were IMUs secured to minimise motion artefacts?
Were there calibration steps to ensure consistent sensor alignment between test sessions?
Did skin movement affect sensor readings?
Consider including a short discussion on how errors were controlled in repeated trials.
The head motion metric (nRMS in cranio-caudal direction) showed higher variability.
Could unintended head movements (e.g., voluntary gaze shifts, trunk compensations) affect reliability?
Consider discussing potential confounders affecting head motion data consistency.
Why was the 10-meter walking test selected?
Could different distances or walking speeds influence reliability?
Would a longer walking test (e.g., 20m) provide more stable gait metrics?
Statical Analysis
Clearly state which outcome measures were normally distributed and which required non-parametric testing.
Define effect sizes (small, moderate, large) according to Cohen’s guidelines.
Could you specify the type of ICC model used (e.g., ICC(3,1)) and provide justification?
Explain the clinical relevance of MDC values, ensuring readers understand what changes matter in rehabilitation.
Discussion
The discussion should explain how these findings translate to real-world gait assessments.
How can rehab professionals use these results to monitor functional improvements in TBI patients?
Should clinicians prioritise certain gait metrics over others when using IMUs in a rehab setting?
The study acknowledges a small sample size, but other limitations should be mentioned, including:
Use of a single walking speed (does reliability change with faster/slower walking?
Limited generalizability beyond a lab setting.
Potential effects of fatigue, cognitive function, or attention on gait metrics.
Suggest longitudinal studies to assess whether these gait metrics change with rehabilitation progress.
Investigate IMU reliability in different environments (e.g., home-based settings and outdoor terrain).
Conclusion
Consider adding a direct clinical takeaway (e.g., "These findings support the integration of wearable sensor-based gait assessments into routine rehabilitation for sTBI patients.").
Comments on the Quality of English Language
Good.
Author Response
Cover letter after revision: Reviewer 2
We would like to thank the Editor and Reviewers for their positive and constructive comments, and for giving us the opportunity of improving and resubmitting our manuscript. We have carefully addressed all the reviewers’ comments.
Abstract.
While the findings are relevant, it would be beneficial to explicitly state how the results can be used in clinical practice.
Response: We would like to thank the Reviewer for having pointed out this theme. The clinical implications of our findings have been explicitly stated in the revised abstract, accordingly.
Specifically, we wrote:
“These findings underscore the reliability and sensitivity of specific IMU-derived indices in detecting meaningful gait changes in sTBI survivors, paving the way for refined assessments and monitoring the rehabilitation process of sTBI survivors.”
Please include a brief mention of ICC values or MDC values for primary gait metrics.
Response: Thank you for your observation. We added the most relevant metric values into the abstract, according to your suggestion.
Introduction.
The second paragraph is too long. Please correct the flow.
Response: Thank you for your comment, we have edited the paragraph by reducing the number of sentences.
Specifically, we wrote:
“Rehabilitation of sTBI patients requires a multidisciplinary approach to enhance recovery and functional independence [5]. Key strategies include physiotherapy for mobility and balance, occupational therapy for daily activities, and rehabilitation for cognitive functions [6]. Emerging interventions, such as virtual reality and robotic-assisted therapies, are gaining attention for their potential benefits [7,8]. However, due to the heterogeneity of TBI presentations, personalized treatment programs remain essential, emphasizing the need for accurate assessment tools [7-9]. Postural stability and motor impairments are critical challenges in sTBI recovery [10-13], making their assessment crucial for understanding limitations and monitoring rehabilitation effectiveness [9].
Wearable inertial measurement units (IMUs) have revolutionized gait assessment by providing objective, portable, and accessible alternatives to traditional methods [14,15]. These devices measure kinematic and spatio-temporal gait parameters, offering insights into stability, symmetry, and movement smoothness. Key metrics, such as normalized Root Mean Square (nRMS) for stability, improved Harmonic Ratio (iHR) for symmetry, and Log Dimensionless Jerk (LDLJ) for smoothness, have demonstrated clinical relevance in sTBI gait assessment [9-11]. Their integration with machine learning models further enhances diagnostic precision [17-19].
Despite their growing use, the reliability and Minimal Detectable Change (MDC) of IMU-derived metrics require further validation. MDC represents the smallest measurable change beyond random variation, crucial for interpreting results and optimizing rehabilitation strategies [20,21]. While nRMS, iHR, and LDLJ have been explored in neurological populations [22-24], systematic studies on their test-retest reliability and MDC in sTBI remain scarce [25]. Addressing this gap is essential for advancing evidence-based clinical decision-making, improving sensitivity in tracking rehabilitation progress, and standardizing wearable sensor-based gait assessments [26].”
L61-63: Reference missing.
Response: Thank you. We added two citations to support the arguments provided.
Methods: How were IMUs secured to minimize motion artifacts? Were there calibration steps to ensure consistent sensor alignment between test sessions? Did skin movement affect sensor readings?
Response: Thanks, we have better clarified details on IMU attachment methods, calibration procedures, and efforts to minimize motion artifacts. IMUs were securely fastened using Velcro straps to minimize motion artifacts and mitigate skin movement. This is the common procedure used in most of the studies investigating gait analysis through IMUs to mitigate skin movements artifacts. We also modified Figure 1 to add a sample of the IMU’s signal from an helthy subject and a sTBI participant. As regards the calibration steps, in the previous version of the manuscript, we stated: “In the static phase at the beginning of each trial (from 3 to 5 s), the gyroscope static bias was removed. Then, to guarantee identical starting conditions for all IMUs located on the upper body, a reference system aligned with the gravity vector was established for each IMU using acceleration data. The rotational matrix between each IMU and this reference system was calculated and applied to the dynamic phase, aligning accelerometer and gyroscope data to approximate the anatomical axes: anteroposterior (AP), mediolateral (ML), and craniocaudal (CC). Gravity was then removed from the CC axis of accelerometer data” Moreover, the calibration procedure embedded into the “Motion Studio” software was applied at the start of each acquisition session. We added this information in the revised “Data collection” paragraph.
Specifically, we wrote:
“Before each acquisition session, a calibration procedure was implemented to ensure consistent sensor alignment between test sessions using the “Motion Studio” software (ADPM, Portland, USA)”.
The head motion metric (nRMS in cranio-caudal direction) showed higher variability. Could unintended head movements (e.g., voluntary gaze shifts, trunk compensations) affect reliability?
Thank you for raising this point. We shortly mentioned this hypothesis in the previous version of the manuscript, now we added a short paragraph in the discussion section to better highlight this relevant aspect:
“As an example, the metric assessing head movement (nRMS in the cranio-caudal direction) showed greater variability. This could be influenced by unintentional factors, such as voluntary eye movements or compensatory trunk adjustments, potentially impacting measurement reliability. Considering the natural tendency of individuals to make subtle postural adaptations during assessment, these secondary movements may introduce variability that is difficult to control, highlighting the need for strategies to minimize their influence in future studies”
Why was the 10-meter walking test selected? Could different distances or walking speeds influence reliability?
The 10-meter walking test was selected due to its feasibility in clinical settings and prior validation for assessing gait parameters in neurological populations. While longer walking tests may provide additional data, shorter distances ensure accessibility for patients with mobility limitations. Of course, further studies could evaluate the reliability of these parameters also during other gait tests.
Statistical Analysis: Clearly state which outcome measures were normally distributed and which required non-parametric testing. Define effect sizes (small, moderate, large) according to Cohen’s guidelines. Specify the type of ICC model used and provide justification.
Thank you for these valuable points. According to your suggestion, at the beginning of the results section we stated:
“The only variable which resulted not normally distributed was the “months from trauma”.
In the methods section, “statistical analysis” paragraph, we explained:
““d” values around 0.2 suggest a small effect, 0.5 a medium effect, and 0.8 or higher a large effect”
And we better detailed the ICC analysis as follows:
“To assess the test-retest reliability, the intra-class correlation coefficient (ICC 3;1) was calculated using a two-way mixed-effects model for absolute agreement within its 95% confidence interval (CI). This choice is justified by the fact that the same raters evaluated all participants, and we aimed to assess the consistency of their ratings rather than generalizing the results to other potential raters”
Discussion: The discussion should explain how these findings translate to real-world gait assessments. How can rehab professionals use these results? Should clinicians prioritize certain gait metrics over others?
Response: Thank you for this suggestion we have better clarified this point
“These results highlight the potential of IMU-based gait assessment in clinical rehabilitation, enabling therapists to monitor recovery progress and tailor interventions based on quantitative gait metrics. Prioritizing stable parameters such as nRMS and iHR may enhance clinical decision-making for sTBI patients”
Limitations: Other limitations should be mentioned, such as use of a single walking speed, limited generalizability, and potential effects of fatigue and cognitive function.
Response: Thanks, we have added these details in the limitation section.
Specifically, we wrote:
“Additionally, the use of a specific evaluation protocol and a single type of sensor may constrain the generalizability of the findings. For example, using only self-selected speed may limit the study's results' applicability to walks of varying speeds.”
And:
“The findings are based on a controlled clinical environment, limiting generalizability to real-world settings. Factors such as fatigue and cognitive function could also influence gait stability and should be investigated in future studies”
Conclusion: Consider adding a direct clinical takeaway.
Response: Thank you, we have added direct clinical takeaway, accordingly.
Specifically, we wrote:
“These results highlight the potential of IMU-based gait assessment in clinical rehabilitation, enabling therapists to monitor recovery progress and tailor interventions based on quantitative gait metrics.”
Round 2
Reviewer 1 Report
Comments and Suggestions for Authors
I have checked the revised manuscript and the authors' reply. It can be accepted now for publication.
Author Response
Response to Reviewer 1.
We would like to acknowledge the reviewer for his/her suggestions that significantly improved our manuscript.
Reviewer 2 Report
Comments and Suggestions for Authors
Abstract
Add more quantitative details in the abstract (e.g., exact reliability coefficients).
Introduction
The flow can be improved by restructuring:
- Explain why assessing gait in TBI is important.
- Introduce IMUs and their role in gait analysis.
- State the research gap and objectives.
- Some sentences are repetitive when discussing IMU applications.
- The research question is not clearly formulated at the end of the section.
Materials and Methods
- The inclusion/exclusion criteria should be structured separately for clarity.
- The IMU placement section can be streamlined by presenting it stepwise.
- The data processing and filtering details should be explicitly stated (e.g., cutoff frequencies for filtering acceleration data).
- The statistical tests should be justified (why was ICC chosen over other reliability metrics?).
- Use subheadings for each method (e.g., Participants, IMU Placement, Data Collection, Statistical Analysis).
- Justify the selection of ICC and MDC calculations.
- Clearly state how missing data were handled.
Results
- Some descriptive statistics are missing (e.g., participant demographics should be summarised numerically).
- The presentation of ICC and MDC results can be structured better.
- The figures need clearer labeling (axes, units, and legends).
Discussion
- The main findings should be summarised first before comparison to prior studies.
- Some comparisons lack depth—more details are needed on how these results compare to previous IMU-based gait studies.
- The clinical relevance should be better explained (How do these findings affect rehabilitation strategies?).
- Limitations section should be expanded to include potential biases (e.g., measurement errors due to sensor drift).
Conclusion
Add a statement on how these findings may inform future research or clinical practice.
General Comments
Lines 14-15: Please use correct references or change the sentence.
Replace "craniocaudal" with "longitudinal" axis throughout.
L245-246: Reference is missing.
Some sentences are too long and should be split for readability.
Please avoid redundancy, especially in the Introduction and Discussion.
Some technical terms should be defined upon first use.
Comments on the Quality of English Language
Good.
Author Response
Response to reviewer 2.
Comment 1. Abstract. Add more quantitative details in the abstract (e.g., exact reliability coefficients).
Response: Exact ICC values had already been added in the revised version after revision round 1. Therefore, we did not add further metric results in the abstract section to maintain readability and respect the words limits provided by Sensors journal.
Comments 2. Introduction
The flow can be improved by restructuring:
- Explain why assessing gait in TBI is important.
- Introduce IMUs and their role in gait analysis.
- State the research gap and objectives.
- Some sentences are repetitive when discussing IMU applications.
- The research question is not clearly formulated at the end of the section.
Response:
- In the revised version of the manuscript after round 1, we already explained why assessing gait in TBI is important. Specifically, we added the following sentence: “However, due to the heterogeneity of TBI presentations, personalized treatment programs remain essential, emphasizing the need for accurate assessment tools [7-9]. Postural stability and motor impairments are critical challenges in sTBI recovery [10-13], making their assessment crucial for understanding limitations and monitoring rehabilitation effectiveness [9].”. In this revised version of the manuscript, we modified this sentence to explicitly mention gait.
Specifically, we wrote: “Postural and gait stability and motor impairments are critical challenges in sTBI recovery [10-13], underscoring the importance of precise assessments to identify limitations and monitoring balance and gait rehabilitation effectiveness [9].”
- Following the reviewer’s suggestion in round 1, we shortened the introduction section in the revised manuscript. Specifically, we wrote, in the previous revised version: “Wearable inertial measurement units (IMUs) have revolutionized gait assessment by providing objective, portable, and accessible alternatives to traditional methods [14,15]. These devices measure kinematic and spatio-temporal gait parameters, offering insights into stability, symmetry, and movement smoothness. Key metrics, such as normalized Root Mean Square (nRMS) for stability, improved Harmonic Ratio (iHR) for symmetry, and Log Dimensionless Jerk (LDLJ) for smoothness, have demonstrated clinical relevance in sTBI gait assessment [9-11]. Their integration with machine learning models further enhances diagnostic precision [17-19].” . In this revised version of the manuscript, we modified this paragraph to introduce IMUs further.
Specifically, we wrote, in this further revised version:
“Wearable inertial measurement units (IMUs) have enhanced gait assessment by providing objective, portable, and accessible alternatives to traditional methods [14,15]. Depending on the number of sensors and their placement setup, IMUs, which embed accelerometers, gyroscopes, and magnetometers, are used in gait analysis to retrieve several kinematic and spatio-temporal gait parameters. Moreover, by directly analyzing the acquired signals during gait, IMUs offer insights into stability, symmetry, and movement smoothness. Clinically relevant metrics, such as normalized Root Mean Square (nRMS) for stability, improved Harmonic Ratio (iHR) for symmetry, and Log Dimensionless Jerk (LDLJ) for smoothness, have demonstrated clinical relevance in sTBI gait assessment [9-11]. Their integration with machine learning models further enhances diagnostic precision [17-19].
- Regarding the research gap, objectives, and research question, we modified the previously revised paragraph according to the reviewer’s suggestion.
Specifically, we wrote, in this revised version:
“Despite the increasing adoption of IMU-derived gait metrics, their reliability and Minimal Detectable Change (MDC) require further validation. MDC, which represents the smallest measurable change beyond random variation, is crucial for results interpretation and optimizing rehabilitation strategies [20,21]. While nRMS, iHR, and LDLJ have been studied in neurological populations [22-24], research on their test-retest reliability and MDC in sTBI remain scarce [25]. Addressing this gap is essential for advancing evidence-based clinical decision-making, improving sensitivity in tracking rehabilitation progress, and standardizing wearable sensor-based gait assessments [26].
Furthermore, when calculating IMU-derived indices for linear walking at steady state, a time series of at least 20 strides is typically required [24, 27]. However, this requirement implies a large acquisition space, which is often unavailable in clinical settings, restricting the application of these metrics outside of laboratory contexts.
We hypothesized that IMU-derived gait indexes would result in reliable and sensitive change scores even for shorter gait bouts, such as during a 10-meters walking test (10MWT), with acceptable intra-class correlation coefficients (ICC) and MDC values. This would enhance the nuance of disposable gait assessment tools for monitoring the rehabilitation process of sTBI survivors. Therefore, this study aims to calculate the test-retest reliability and the MDC of the nRMS, iHR, and LDLJ parameters, measured through IMU sensors, in survivors from sTBI.”
Comments 3. Materials and Methods: The inclusion/exclusion criteria should be structured separately for clarity.
Response. Inclusion criteria have been divided into separate subsections, accordingly.
Comment 4. The IMU placement section can be streamlined by presenting it stepwise.
Response. The reference to figure 1, where the experimental setup is clearly described, has been added at the end of the description of sensor placements.
Comment 5. The data processing and filtering details should be explicitly stated (e.g., cutoff frequencies for filtering acceleration data).
Response. Information about data processing and filtering had already been provided in the first version of the manuscript. Specifically, in the Data collection subsection, we stated: “In the static phase at the beginning of each trial (from 3 to 5 s), the gyroscope static bi-as was removed. Then, to guarantee identical starting conditions for all IMUs located on the upper body, a reference system aligned with the gravity vector was established for each IMU using acceleration data. The rotational matrix between each IMU and this reference system was calculated and applied to the dynamic phase, aligning accelerometer and gyroscope data to approximate the anatomical axes: anteroposterior (AP), mediolateral (ML), and craniocaudal (CC). Gravity was then removed from the CC axis of accelerometer data [35]. To filter the data, a second-order Butterworth low-pass filter was applied with cutoff frequencies of 10 Hz for accelerometers and 6 Hz for gyroscopes.”.
Comment 6-7. The statistical tests should be justified (why was ICC chosen over other reliability metrics?) - Justify the selection of ICC and MDC calculations.
Response. The use of ICC as the test-retest reliability measure was already justified in the “Statistical Analysis” subsection of the revised manuscript following round 1. Specifically, we stated: “To assess the test-retest reliability, the intra-class correlation coefficient (ICC 3;1) was calculated using a two-way mixed-effects model for absolute agreement within its 95% confidence interval (CI). This choice is justified by the fact that the same raters evalu-ated all participants, and we aimed to assess the consistency of their ratings rather than generalizing the results to other potential raters. ICC values were interpreted as follows: less than 0.5 as poor reliability; 0.5 - 0.75 as moderate reliability; 0.75 - 0.9 as good reliability; and greater than 0.90 as excellent reliability [43]. Statistical analyses were carried out at 95% significance level.”. The Intraclass Correlation Coefficient (ICC) is the most effective statistical metric for assessing test-retest reliability with continuous data. The ICC measures the consistency or agreement of repeated measurements of the same subject, taking into account both systematic bias and random measurement error. It is a more robust measure of reliability than Pearson's correlation because it accounts for both within- and between-subject variability, and it was better suited to our study's objectives than Bland-Altman analysis because we were evaluating absolute agreement rather than relative consistency. However, we decided not to include these considerations in the revised version of the manuscript because using ICC for our purposes is a well-established methodology and introducing them would have added no novelties to the manuscript (see, for example, the COSMIN statement).
Comment 8. Use subheadings for each method (e.g., Participants, IMU Placement, Data Collection, Statistical Analysis).
Response. The subheadings suggested by the reviewer had already been used in previous versions of the manuscript. As a result, in this revised version, we only added the subject 'Inclusion Criteria' in response to the reviewer's previous comment.
Comment 9. Clearly state how missing data were handled.
Response. No missing data in the dataset were found. This information has been added at the end of the “Statistical Analysis” section. Specifically, we wrote:
“No missing data across the variables and observations were found.”
Comment 10. Results. Some descriptive statistics are missing (e.g., participant demographics should be summarised numerically).
Response. Descriptives demographics had already been numerically summarized within Table 1 in the previous versions of the manuscript.
Comment 11. The presentation of ICC and MDC results can be structured better.
Response. ICC and MDC results were clearly presented in Table 1 and Table 2. Therefore, we did not modify this section further. If the reviewer has more specific suggestions for restructuring the section, we are ready to consider them.
Comment 12. The figures need clearer labeling (axes, units, and legends).
Response. Figure 2 was explained in its figure caption. Specifically, we wrote in the previous version: “Figure 2. Distributions of the test – retest differences within the sample. Combined density, boxplot, and scatterplot representations of normalized root mean square (nRMS) values, improved harmonic ratios (iHR), and log – dimensionless jerk score of accelerations (LDLJa) and angular velocities (LDLJw) across antero – posterior (AP), medio- lateral (ML), and cranio – caudal (CC) directions. Each plot consists of a density plot, indicating the distribution of values, boxplot, summarizing the median and interquartile range, and a scatterplot, displaying individual test – retest difference to reflect underlying variability.” . In this revised version of the figure caption, we added a description of the axes.
Specifically, we wrote: “The x-axes represent the test–retest differences for each respective metric, while the y-axes show the density estimates for the distribution of differences."
Comment 13. Discussion. The main findings should be summarised first before comparison to prior studies.
Response. We summarized the main findings before comparing them to prior studies, accordingly. Specifically, we wrote, in this revised version:
“This study aimed to assess the test – retest reliability and the minimal detectable change scores of a set of IMU–derived indices quantifying gait stability, symmetry, and smoothness, in a population of sTBI survivors as assessed during a 10 meters walking test. The results showed moderate to excellent test-retest reliability for all investigated parameters, with nRMS and iHR showing particularly high reliability in specific directions. Notably, nRMS calculated from pelvic and trunk AP and ML accelerations demonstrated excellent reliability and low MDC values, indicating its suitability for tracking gait stability changes in this population. The iHR also demonstrated high reliability in the AP and CC directions, but its ML component showed greater variability. In contrast, LDLJa and LDLJw had higher test-retest differences and larger MDC val-ues, indicating greater measurement variability. These findings indicate that, while some IMU-based gait parameters offer solid clinical applicability, others require fur-ther refinement to improve reliability in short walking bouts.”
Comment 14. Some comparisons lack depth—more details are needed on how these results compare to previous IMU-based gait studies.
Response. In this revised version of the manuscript, we added further details when comparing our results with previous research, accordingly. Specifically, we wrote:
“This is consistent with previous research on the reliability of acceleration – derived RMS in healthy adults performing a 10 meters walking test [47,48], where ICC values of about 0.80 for RMS were reported [48]. These findings suggest that nRMS can be considered as reliable in TBI survivors even for short walking bouts”
And:
“On the other hand, the iHR ML component exhibited larger MDC values, suggesting a reduced sensitivity to slight changes. This is in line with previous research on healthy subjects, where the iHR ML showed substantial between-subject variability [52].”
Comment 15. The clinical relevance should be better explained (How do these findings affect rehabilitation strategies?).
Response. In this revised version of the manuscript, we further explained how our results may affect rehabilitation strategies.
Specifically, we wrote: “These results highlight the potential of IMU-based gait assessment in clinical rehabilitation, enabling therapists to monitor recovery progress and tailor interventions based on quantitative gait metrics. Prioritizing stable parameters such as nRMS and iHR may enhance clinical decision-making for sTBI patients by focusing gait rehabilitation programs on gait stability and trunk symmetry.”
Comment 16. Limitations section should be expanded to include potential biases (e.g., measurement errors due to sensor drift).
Response. The limitations paragraph has been expanded to include the potential bias due to the acceleration drift.
Specifically, we wrote: “Furthermore, while the calibration procedure was performed prior to each walking trial using the embedded procedure of the acquisition software, and the short walking bout makes the procedure quite safe in terms of calibration stability, we cannot completely rule out errors caused by acceleration drift during the procedure”.
Comment 17. Conclusion. Add a statement on how these findings may inform future research or clinical practice.
Response. In this revised version of the manuscript, we improved the statement about how our findings may inform future research or clinical practice, accordingly.
Specifically, we wrote:
“These results highlight the potential of IMU-based gait assessment in clinical rehabilitation, enabling therapists to monitor recovery progress and tailor interventions based on quantitative gait metrics. Clinicians can plan gait rehabilitation interventions for TBI survivors that focus on trunk stability and symmetry by assessing RMS and iHR from IMUs. Further research is required to refine these indices and explore their reliability across different populations with gait impairments and under varying acquisition protocols, as well as their responsiveness to rehabilitation. “
Comment 18. General Comments. Lines 14-15: Please use correct references or change the sentence.
Response. Lines 14-15 refer to the Abstract section, where no references are allowed. So, unfortunately, we cannot address this comment.
Comment 19. Replace "craniocaudal" with "longitudinal" axis throughout.
Response. We agree with the reviewer that in terms of reference systems, craniocaudal represents the longitudinal axis. However, the term craniocaudal for the longitudinal axis has previously been used in several studies evaluating IMU-derived metrics. Since this work is also intended for clinical readers, we prefer not to change this term throughout the manuscript. However, when we first mentioned the CC axis, we clarified that it refers to the longitudinal axis.
Specifically, we wrote, in the “Data Collection” subsection of the Materials and Methods section:
“The rotational matrix between each IMU and this reference system was calculated and applied to the dynamic phase, aligning accelerometer and gyroscope data to approxi-mate the anatomical axes: anteroposterior (AP), mediolateral (ML), and craniocaudal (CC), with the latter referring to the longitudinal axis.”
Comment 20. L245-246: Reference is missing.
Response. The reference for Cohen’s d interpretation has been added, accordingly.
Comment 21-22. Some sentences are too long and should be split for readability. - Please avoid redundancy, especially in the Introduction and Discussion.
Response. We modified the introduction and discussion sections to improve readability and avoid redundancies, accordingly.
Comment 23. Some technical terms should be defined upon first use.
Response. We checked the manuscript and did not find acronyms that were not spelled out before their first use. If the reviewer has more specific suggestions, we are ready to consider them.
